**www.cambridge.org/qrd**

## Research Article

Convert hydrophobic alpha-helix to hydrophilic alpha-helix:Protein engineering; QTY code; Water-soluble transmembrane protein megacomplex

**Corresponding author:**
Shuguang Zhang;
Email: Shuguang@mit.edu

# Structural bioinformatic study of human mitochondrial respiratory integral membrane megacomplex and its AlphaFold3 predicted water-soluble QTY megacomplex analog

Edward Chen[1] [ID] and Shuguang Zhang[2] [ID]

[1]Independent Researcher, Pittsburgh, PA, USA and [2]Media Lab, Massachusetts Institute of Technology, Cambridge, MA, 02139, USA

## Abstract

Human mitochondrial Complex I is one of the largest multi-subunit membrane protein megacomplexes, which plays a critical role in oxidative phosphorylation and ATP production. It is also involved in many neurodegenerative diseases. However, studying its structure and the mechanisms underlying proton translocation remains challenging due to the hydrophobic nature of its transmembrane parts. In this structural bioinformatic study, we used the QTY code to reduce the hydrophobicity of megacomplex I, while preserving its structure and function. We carried out the structural bioinformatics analysis of 20 key enzymes in the integral membrane parts. We compare their native structure, experimentally determined using Cryo-electron microscopy (CryoEM), with their water-soluble QTY analogs predicted using AlphaFold 3. Leveraging Alpha-Fold 3's advanced capabilities in predicting protein–protein complex interactions, we further explore whether the QTY-code integral membrane proteins maintain their protein–protein interactions necessary to form the functional megacomplex. Our structural bioinformatics analysis not only demonstrates the feasibility of engineering water-soluble integral membrane proteins using the QTY code, but also highlights the potential to use the water-soluble membrane protein QTY analogs as soluble antigens for discovery of therapeutic monoclonal antibodies, thus offering promising implications for the treatment of various neurodegenerative diseases.

## Introduction

The mitochondrial megacomplex produces most of the energy in the human body (Stroud et al., 2016; Wu et al 2016; Guo et al., 2017). The respiratory chain complexes (RCCs) include Complex I (CI), Complex II (CII), Complex III (CIII), and Complex IV (CIV) that are located in the inner mitochondrial membrane are critical in energy conversion. Complex I (NADH: ubiquinone oxidoreductase) is the entry point for electrons to enter the RCCs, where two electrons from NADH are catalyzed into quinone (Berrisford and Sazanov, 2009; Efremov and Sazanov, 2012; Hirst, 2013). Then, Complex I, Complex III (NADH: CIII, cytochrome bc1 complex), and Complex IV (NADH: cytochrome c oxidase) couples electron transfer by using the reduced potential of NADH to drive four protons across the inner membrane, leading to ATP synthesis in Complex V (CV) (Berrisford and Sazanov, 2009; Hirst, 2013; Vinothkumar et al., 2014; Zickermann et al., 2015; Zhu et al., 2016).

The L-shaped Complex I enzyme is one of the largest multi-subunit membrane protein complexes with 45 subunits (Mimaki et al., 2012; Stroud et al., 2016; Wu et al., 2016; Zhu et al., 2016) split into three modules (Efremov et al., 2010; Wirth et al., 2016). The NADH oxidation module (N module) and ubiquinone (Q) reduction module (Q module) form the peripheral arm, and the proximal and distal proton translocation module ($P_P$ and $P_D$ modules) form the membrane arm (Sharma et al., 2009; Parey et al., 2021). The hydrophobic transmembrane arm or the P module containing the mtDNA-encoded subunits is embedded in the inner mitochondrial membrane, where the subunits are stabilized by tightly bound lipids (Fiedorczuk et al., 2016). The transmembrane arm includes 3 highly hydrophobic subunits of ND2, ND4, and ND5, which contains around 15 transmembrane domains (Mimaki et al., 2012). The three antiporter-like subunits located inside the membrane arm are largely responsible for proton pumping activities.

With Complex I being an integral part of the RCCs, the dysfunction of the complex impairs oxidative phosphorylation and reduces ATP synthesis. These impairments prevent metabolic processes and lead to diseases including Alzheimer's and Parkinson's diseases, Friedreich's ataxia, amyotrophic lateral sclerosis, Hurthle cell thyroid carcinoma, Leber's hereditary optic neuropathy, Leigh syndrome, and so forth (Distelmaier et al., 2009; Guo et al., 2017; McGregor et al., 2023; Menezes et al., 2014; Rodenburg, 2016; Sharma et al., 2009). In addition, Complex I has

been linked as a major source of reactive oxygen species, which could damage mitochondria DNA and lead to aging.

Our study focuses on the 20 inner membrane proteins of the Complex I membrane that have direct medical relevance, including NDUA1, NDUA3, NDUAB, NDUAD, NDUB1, NDUB3, NDUB4, NDUB5, NDUB6, NDUB8, NDUBB, NDUC1, NDUC2, NU1M, NU2M, NU3M, NU4M, NU5M, NU6M, and NU4LM (Table 1). The other non-membrane proteins in the megacomplex are not subjected to the current study.

Traditionally, researchers use X-ray crystallography and NMR spectroscopy to study protein structures. Recently, high-resolution cryo-electron microscopy (CryoEM) has become the mainstream method used to study protein structures at near-atomic resolution by freezing the target specimen at temperatures of liquid nitrogen or nitrogen helium (Henderson et al., 2011; Milne et al., 2013; Vinothkumar and Henderson, 2016). In our study, our baseline native structure is from the CryoEM structure megacomplex at 3.70Å resolution (Guo et al., 2017).

However, despite these advancements, studying the structure and functions of these multi-subunit membrane proteins remains challenging due to the need of detergent for solubilization after isolating the proteins from the hydrophobic transmembrane regions. This process is often complicated and time-consuming before obtaining a high-resolution structure elucidation (Carpenter et al., 2008; Vinothkumar and Henderson, 2010).

Current efforts to solubilize proteins include ProteinMPNN, which utilizes message-passing neural networks to predict and design the amino acid sequence that would fold into the desired shape. ProteinMPNN yields better results in predicting the hydrophobic amino acids for a protein backbone compared to Rosetta (Dauparas et al., 2022). Recently, researchers built on top of ProteinMPNN to devise SolubleMPNN trained on only soluble proteins, which was applied to engineer soluble variants of bacteriorhodopsin, successfully converting a membrane protein into a soluble one, while maintaining its core function and ligand-binding ability (Nikolaev et al., 2024). A generalization approach for the computational design of soluble membrane proteins was also explored by using ProteinMPNN on AlphaFold 2-generated structures, which generated soluble analogs for both rhomboid protease fold and seven-helix GPCR fold (Goverde et al., 2024).

Instead of taking a computational approach, we applied the QTY code to systematically engineer water-soluble analogs with reduced hydrophobicity in membrane proteins. The QTY concept was inspired by high-resolution (1.5Å) electron density maps, which revealed structural similarities between hydrophobic and polar amino acids leucine (L) *vs* glutamine (Q); isoleucine (I)/valine (V) *vs* threonine (T); and phenylalanine (F) *vs* tyrosine (Y) (Zhang et al., 2018; Tegler et al., 2020; Zhang and Egli, 2022). In our previous experiments, using the simple and straightforward QTY code, we successfully bioengineered detergent-free chemokine (Zhang et al., 2018; Qing et al., 2019; Tegler et al., 2020), cytokine receptors (Hao et al., 2020) and bacterial histidine kinase (Li et al., 2024). After these detergent-free membrane proteins were expressed and purified, these QTY analogs demonstrated structural stability, retained their ligand-binding capabilities, and intact four enzymatic activities, making them ideal

**Table 1.** The protein names, UniProt ID, and CryoEM structure (Å) with PBD ID

| Name (Uniprot ID) | Structure (Å, PDB ID) | Tissue expression | Medical relevance |
|---|---|---|---|
| NDUAI (O15239) | CryoEM(3.70Å, 5XTC) | Heart, skeletal muscle | Oxidative phosphorylation |
| NDUA3 (O95167) | CryoEM(3.70 Å, 5XTC) | Heart, kidney, liver | Electron transport |
| NDUAB (Q86Y39) | CryoEM(3.70 Å, 5XTC) | Heart, brain, pancreas | ATP production |
| NDUAD (Q9P0J0) | CryoEM(3.40 Å, 5XTB) | Liver, kidney, placenta | Cell death regulation |
| NDUB1 (O75438) | CryoEM(3.70 Å, 5XTC) | Heart, skeletal muscle | Electron transport |
| NDUB3 (O43676) | CryoEM(3.70 Å, 5XTC) | Heart, kidney, liver | ATP production |
| NDUB4 (O95168) | CryoEM(3.70 Å, 5XTC) | Heart, brain, kidney | Oxidative phosphorylation |
| NDUB5 (O43674) | CryoEM(3.70 Å, 5XTC) | Heat, skeletal muscle | ATP production |
| NDUB6 (O95139) | CryoEM(3.70 Å, 5XTC) | Heart, brain, liver | Electron transport |
| NDUB8 (O95169) | CryoEM(3.70 Å, 5XTC) | Heart, muscle, liver | Mitochondrial protein import |
| NDUBB (Q9NX14) | CryoEM(3.70 Å, 5XTC) | Heat, skeletal muscle | Oxidative phosphorylation |
| NDUC1 (O43677) | CryoEM(3.70 Å, 5XTC) | Heart, tongue, kidney | Electron transport |
| NDUC2 (O95298) | CryoEM(3.70 Å, 5XTC) | Adrenal gland, heart | Neutrophil degranulation |
| NUIM (P03886) | CryoEM(3.70 Å, 5XTC) | Heart, skeletal muscle | Mitochondrial protein degradation |
| NU2M (P03891) | CryoEM(3.70 Å, 5XTC) | Heart, brain, liver | Aerobic respiration |
| NU3M (P03897) | CryoEM(3.70 Å, 5XTC) | Uterine tube, kidney | ATP synthesis |
| NU4M (P03905) | CryoEM(3.70 Å, 5XTC) | Uterine tube, brain | Cerebellum development |
| NU5M (P03915) | CryoEM(3.70 Å, 5XTC) | Ubiquitous | Electron transport |
| Nu6M (P03923) | CryoEM(3.70 Å, 5XTC) | Mucosa of stomach | Electron transport |
| NU4LM (P03901) | CryoEM(3.70 Å, 5XTC) | Visual cortex, tongue | Proton translocation |

*Note:* The lists of tissue location, medical relevance, and function are not exhaustive. Updated results become available from more and more recent studies.

candidates for further studies and use as antigens to generate therapeutic monoclonal antibodies (mAbs).

Google's DeepMind released the breakthrough AlphaFold 2 in 2021 (Jumper et al., 2021; Jumper and Hassabis, 2022), and it placed over 214 million AlphaFold 2 predicted protein structures at the European Bioinformatic Institute (EBI) (Tunyasuvunakool et al., 2021). We previously used AlphaFold 2 to predict membrane protein QTY analog protein structures. The QTY code was applied to 7 chemokine receptors (Skuhersky et al., 2021), human olfactory receptors (Johnsson et al., 2024), glucose transporters (Smorodina et al., 2022b), solute carrier transporters (Smorodina et al., 2022a), ABC transporters (Pan et al., 2024), and neurological transporters including serotonin, norepinephrine, dopamine transporters (Karagöl et al., 2024) and another synaptic vesicle protein subgroup of glutamate transporters (VGLUTs) (Karagöl et al., 2024). We also designed reverse QTY analogs of human serum albumin to effectively facilitate the release of antitumor drugs in mice (Meng et al., 2023). The water-soluble chemokine receptor CXCR4$^{QTY}$ analog has been successfully used in biomimetic sensors (Qing et al., 2023). We also used AlphaFold 2 to predict QTY analogs of beta-sheet-rich antibody IgG (Li et al., 2023) and bacterial beta-barrel proteins (Sajeev-Sheeja et al., 2023) and beta-barrel enzymes (Sajeev-Sheeja and Zhang, 2024).

In May 2024, AlphaFold was upgraded to version 3 as AlphaFold 3, featuring an enhanced diffusion-based architecture that enables accurate prediction of multiple structures of protein complexes. Additionally, AlphaFold 3 extends its capabilities beyond protein structure prediction to include DNA, RNA, and small molecules including ligands and other proteins (Abramson et al., 2024). Notably, it can model interactions between odorants and the human olfactory receptor OR1A2, as well as spermidine with the trace amine receptor TAAR9 (Johnsson et al., 2024).

To build on top of our previous studies and utilize AlphaFold 3's advanced capabilities, we used AlphaFold 3 to test the structural stability of the QTY analog megacomplex of the human mitochondrial respiratory system. In addition, we conducted bioinformatic studies using AlphaFold 3 to predict the protein–protein interactions of QTY analogs compared to their native structures. Here, we report the structural bioinformatic studies of experimentally determined Complex I and its AlphaFold 3-predicted water-soluble QTY analog. We also provide the superpositions of native and QTY analog proteins, their surface hydrophobicity analyses, and finally the protein–protein interaction analyses of the hydrophobic native Complex I megacomplex and their hydrophilic QTY analogs.

## Results and discussion

### The rationale of the QTY Code

The hydrophobic nature of the membrane proteins makes it challenging to study their structure and function. We asked if it is possible to systematically exchange the hydrophobic amino acids into hydrophilic ones to make these membrane proteins more water-soluble. Indeed, the structural similarities between the electron density maps of Q and L, T and V/I, and Y and F make it possible to systematically replace the hydrophobic amino acids with hydrophilic ones: leucine (L) with glutamine (Q), isoleucine (I) and valine (V) with threonine (T), and phenylalanine (F) with tyrosine (Y). While bringing changes to protein sequence and amino acid composition, the QTY analogs demonstrate reduced hydrophobic surfaces and exhibit similar isoelectric points (pI) and molecular weights (MW) when compared to the native transmembrane proteins (Table 2).

### Protein sequence alignments and other characteristics

The protein sequences of the twenty mitochondrial proteins are aligned with their QTY analogs (Figure 1). The QTY substitution of the twenty proteins resulted in overall changes to their amino acid composition from 4.90% to 37.36% and changes in the transmembrane domain from 26.09% to 66.67%. Despite the changes to the structure and composition, the pI only changed slightly due to the neutral charges of Q (glutamine), T (threonine), and Y (tyrosine). Thus, the substitutions introduced by the QTY code do not add any basic or acidic amino acids. The MW of the proteins increased slightly due to the replacement of leucine (L: 131.17 Da) *vs* glutamine (Q: 146.14 Da), isoleucine (I: 131.17 Da), valine (V: 117.15 Da) *vs* threonine (T: 119.12 Da), and phenylalanine (F: 165.19 Da) *vs* tyrosine (Y: 181.19 Da).

### Superpositions of native CryoEM transmembrane enzymes and their water-soluble QTY analogs

We asked if the molecular structure of the twenty proteins in the mitochondrial Complex I is similar to their QTY analogs after applying the QTY substitution (Figure 2). The native structures of the mitochondrial complex are determined experimentally using CryoEM (PDB: 5XTC). The structures of the QTY analogs are predicted using AlphaFold 3. The superpositions of the transmembrane enzymes and their respective QTY analogs are: NDUA1 *vs* NDUA1$^{QTY}$, NDUA3 *vs* NDUA3$^{QTY}$, NDUAB *vs* NDUAB$^{QTY}$, NDUAD *vs* NDUAD$^{QTY}$, NDUB1 *vs* NDUB1$^{QTY}$, NDUB3 *vs* NDUB3$^{QTY}$, NDUB4 *vs* NDUB4$^{QTY}$, NDUB5 *vs* NDUB5$^{QTY}$, NDUB6 *vs* NDUB6$^{QTY}$, NDUB8 *vs* NDUB8$^{QTY}$, NDUBB *vs* NDUBB$^{QTY}$, NDUC1 *vs* NDUC1$^{QTY}$, NDUC2 *vs* NDUC2$^{QTY}$, NU1M *vs* NU1M$^{QTY}$, NU2M *vs* NU2M$^{QTY}$, NU3M *vs* NU3M$^{QTY}$, NU4M *vs* NU4M$^{QTY}$, NU5M *vs* NU5M$^{QTY}$, NU6M *vs* NU6M$^{QTY}$, and NU4LM *vs* NU4LM$^{QTY}$ (Figure 2).

The structures of the native mitochondrial proteins superposed well with their QTY analogs, with root mean square deviation (RMSD) ranging from 0.315Å to 1.302Å with one exception of NDUB1, which has a slightly higher RMSD of 2.316Å (Table 2). Overall, the low RMSD indicates both the capability of AlphaFold 3 in predicting the structures of novel protein designs and the minimal structural change in the QTY analogs compared to their native counterparts.

### Superpositions of AlphaFold 3-predicted native transmembrane enzymes and their water-soluble QTY analogs

We also ask how well the AlphaFold 3-predicted mitochondrial membrane proteins superpose with their QTY analogs (Figure 3). The structures superposed very well with low RMSD (Figure 3): a) NDUA1 *vs* NDUA1$^{QTY}$ (RMSD = 0.637Å), b) NDUA3 *vs* NDUA3$^{QTY}$ (RMSD = 0.400Å), c) NDUAB *vs* NDUAB$^{QTY}$ (RMSD = 0.374Å), d) NDUAD *vs* NDUAD$^{QTY}$ (RMSD = 0.570Å), e) NDUB1 *vs* NDUB1$^{QTY}$ (RMSD = 2.180Å), f) NDUB3 *vs* NDUB3$^{QTY}$ (RMSD = 1.110Å), g) NDUB4 *vs* NDUB4$^{QTY}$ (RMSD = 0.687Å), h) NDUB5 *vs* NDUB5$^{QTY}$ (RMSD = 0.511Å), i) NDUB6 *vs* NDUB6$^{QTY}$ (RMSD = 3.127Å), j) NDUB8 *vs* NDUB8$^{QTY}$ (RMSD = 0.773Å), k) NDUBB *vs* NDUBB$^{QTY}$ (RMSD = 0.478Å), l) NDUC1 *vs* NDUC1$^{QTY}$ (RMSD = 1.283Å), m) NDUC2 *vs* NDUC2$^{QTY}$ (RMSD = 0.184Å), n) NU1M *vs* NU1M$^{QTY}$ (RMSD = 0.308Å), o) NU2M *vs* NU2M$^{QTY}$ (RMSD = 0.390Å), p) NU3M *vs* NU3M$^{QTY}$ (RMSD = 0.837Å), q) NU4M *vs* NU4M$^{QTY}$ (RMSD = 0.270Å), r) NU5M *vs* NU5M$^{QTY}$ (RMSD =

**Table 2.** The characteristics of integral membrane protein enzymes and their QTY analogs

| Name | RMSD (Å) | pI | Mw (kDw) | TM variation (%) | Overall variation (%) |
|---|---|---|---|---|---|
| NDUA1 CryoEM | – | 8.93 | 8.07 | – | – |
| NDUA1 QTY | 0.600 Å | 8.88 | 8.14 | 47.62 | 14.29 |
| NDUA3 CryoEM | | 8.26 | 9.15 | – | – |
| NDUA3 QTY | 0.605 Å | 8.22 | 9.24 | 57.14 | 14.46 |
| NDUAB CryoEM | | 8.95 | 14.72 | – | – |
| NDUAB QTY | 0.474 Å | 8.91 | 14.78 | 26.09 | 8.57 |
| NDUAD CryoEM | | 8.23 | 16.57 | – | – |
| NDUAD QTY | 0.544 Å | 8.23 | 16.56 | 31.82 | 4.90 |
| NDUB1 CryoEM | | 9.04 | 6.83 | – | – |
| NDUB1 QTY | 2.316 Å | 8.98 | 6.87 | 47.06 | 14.04 |
| NDUB3 CryoEM | | 9.19 | 11.27 | – | – |
| NDUB3 QTY | 0.729 Å | 9.05 | 11.37 | 43.48 | 10.31 |
| NDUB4 CryoEM | | 9.84 | 15.08 | – | – |
| NDUB4 QTY | 0.599 Å | 9.77 | 15.11 | 50.00 | 7.03 |
| NDUB5 CryoEM | | 6.95 | 17.00 | – | – |
| NDUB5 QTY | 1.260 Å | 6.95 | 17.03 | 61.90 | 9.09 |
| NDUB6 CryoEM | | 9.62 | 15.36 | – | – |
| NDUB6 QTY | 0.931 Å | 9.57 | 15.38 | 52.63 | 7.87 |
| NDUB8 CryoEM | | 5.45 | 18.77 | – | – |
| NDUB8 QTY | 0.820 Å | 5.45 | 18.86 | 47.62 | 6.33 |
| NDUBB CryoEM | | 4.62 | 14.31 | – | – |
| NDUBB QTY | 0.699 Å | 4.62 | 14.42 | 66.67 | 11.29 |
| NDUC1 CryoEM | | 9.16 | 5.94 | – | – |
| NDUC1 QTY | 1.302 Å | 9.05 | 6.01 | 52.63 | 20.41 |
| NDUC2 CryoEM | | 9.03 | 14.19 | – | – |
| NDUC2 QTY | 0.315 Å | 8.96 | 14.29 | 45.00 | 7.56 |
| NU1M CryoEM | | 6.11 | 35.66 | – | – |
| NU1M QTY | 0.662 Å | 6.11 | 36.24 | 45.24 | 23.90 |
| NU2M CryoEM | | 9.83 | 38.96 | – | – |
| NU2M QTY | 0.695 Å | 9.57 | 39.56 | 41.90 | 25.36 |
| NU3M CryoEM | | 4.45 | 13.19 | – | – |
| NU3M QTY | 0.583Å | 4.45 | 13.52 | 63.49 | 34.78 |
| NU4M CryoEM | | 9.40 | 51.58 | – | – |
| NU4M QTY | 0.590 Å | 9.19 | 52.37 | 43.53 | 22.00 |
| NU5M CryoEM | | 9.14 | 67.03 | – | – |
| NU5M QTY | 0.560 Å | 8.93 | 68.02 | 44.30 | 23.22 |
| NU6M CryoEM | | 4.17 | 18.62 | – | – |
| NU6M QTY | 0.866 Å | 4.17 | 18.96 | 51.59 | 37.36 |
| NU4LM CryoEM | | 5.72 | 10.74 | – | – |
| NU4LM QTY | 0.619 Å | 5.72 | 10.96 | 46.03 | 29.59 |

*Note:* The twenty membrane proteins are listed in the same order as Figure 1. RMSDs were calculated after missing residuals (unstructured loops) in the native CryoEM-determined structures and the corresponding residuals in the predicted QTY structures were cut out. If the native protein was a dimer, one monomer was also cut out. The QTY amino acid substitutions in the transmembrane (TM) are significant between 26.09% and 66.67%, whereas the overall structural changes are between 4.90% and 37.36%.

Abbreviations: pI, isoelectric focusing; MW, molecular weight; TM, transmembrane; –, not applicable, and RMSD, residue mean square distance.

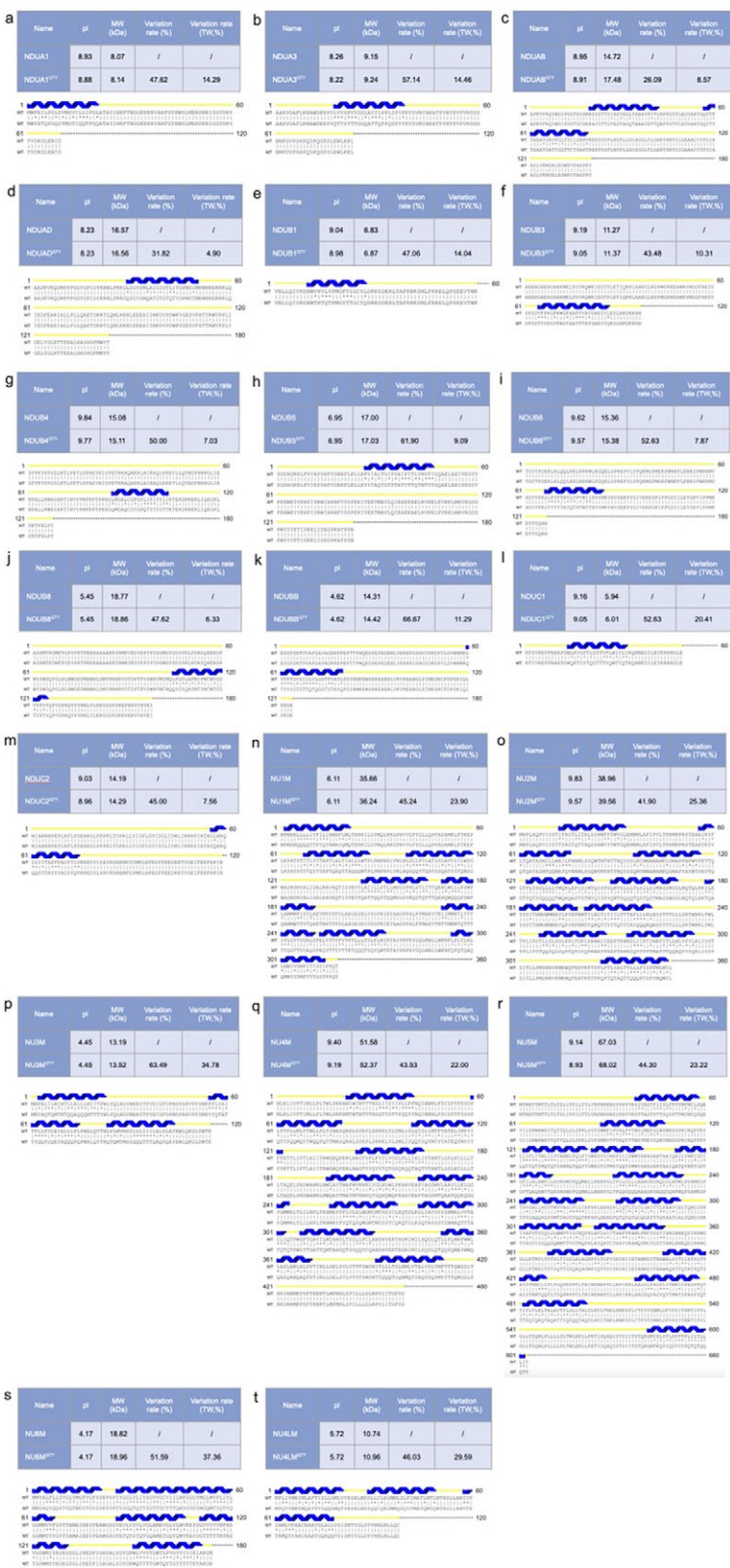

**Figure 1. Protein sequence alignments of twenty integral membrane enzymes with their water-soluble QTY analogs**. The symbols | and * indicate whether amino acids are identical or different, respectively. Please note the Q, T, and Y amino acids (red) replacing L, V, I, and F, respectively. The alpha helices (blue) are shown above the protein sequences. The characteristics of natural and QTY analogs listed are isoelectric focusing (pI), molecular weight (MW), total variation %, and transmembrane variation %. The alignments are: a) NDUA1 vs NDUA1$^{QTY}$, b) NDUA3 vs NDUA3$^{QTY}$, c) NDUAB vs NDUAB$^{QTY}$, d) NDUAD vs NDUAD$^{QTY}$, e) NDUB1 vs NDUB1$^{QTY}$, f) NDUB3 vs NDUB3$^{QTY}$, g) NDUB4 vs NDUB4$^{QTY}$, h) NDUB5 vs NDUB5$^{QTY}$, i) NDUB6 vs NDUB6$^{QTY}$, j) NDUB8 vs NDUB8$^{QTY}$, k) NDUBB vs NDUBB$^{QTY}$, l) NDUC1 vs NDUC1$^{QTY}$, m) NDUC2 vs NDUC2$^{QTY}$, n) NU1M vs NU1M$^{QTY}$, o) NU2M vs NU2M$^{QTY}$, p) NU3M vs NU3M$^{QTY}$, q) NU4M vs NU4M$^{QTY}$, r) NU5M vs NU5M$^{QTY}$, s) NU6M vs NU6M$^{QTY}$, and t) NU4LM vs NU4LM$^{QTY}$. Although there are significant QTY changes in the TM alpha helices (26.09%–66.67%), their changes in MW and pI are insignificant. The protein alignment panels in Figure 1 are too small to visualize. For enlarged individual panels, please see Supplementary Information.

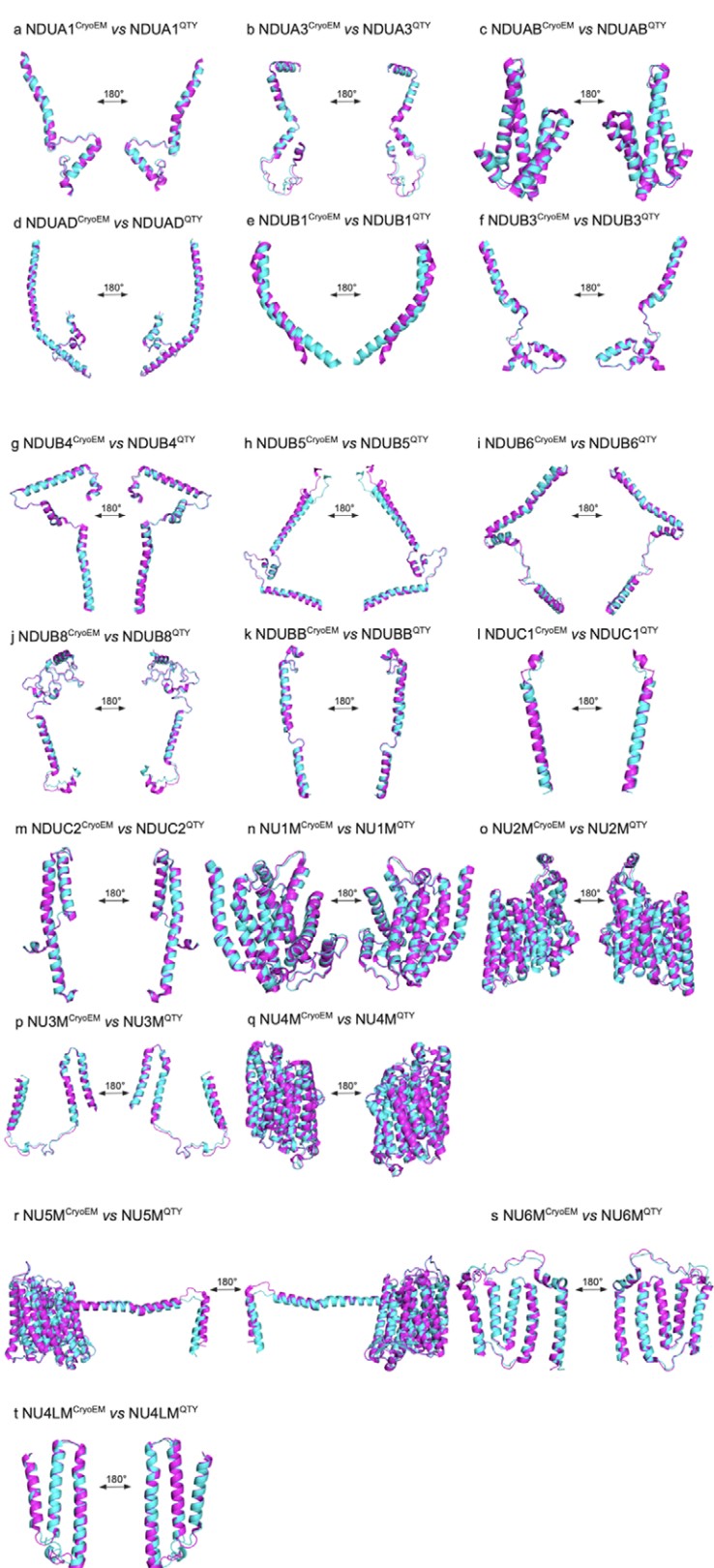

**Figure 2. Superpositions of twenty human CryoEM-determined structures of membrane enzymes and their AlphaFold 3-predicted water-soluble QTY analogs**. The CryoEM-determined structures of the native transporters are obtained from the Protein Data Bank (PDB). The CryoEM structures (magenta) are superposed with their QTY analogs (cyan) predicted by AlphaFold 3. These superposed structures show that the membrane proteins and their QTY analogs have very similar structures. For clarity of direct comparisons, unstructured loops in the CryoEM structures were removed in the QTY analogs. a) NDUA1 vs NDUA1$^{QTY}$, b) NDUA3 vs NDUA3$^{QTY}$, c) NDUAB vs NDUAB$^{QTY}$, d) NDUAD vs NDUAD$^{QTY}$, e) NDUB1 vs NDUB1$^{QTY}$, f) NDUB3 vs NDUB3$^{QTY}$, g) NDUB4 vs NDUB4$^{QTY}$, h) NDUB5 vs NDUB5$^{QTY}$, i) NDUB6 vs NDUB6$^{QTY}$, j) NDUB8 vs NDUB8$^{QTY}$, k) NDUBB vs NDUBB$^{QTY}$, l) NDUC1 vs NDUC1$^{QTY}$, m) NDUC2 vs NDUC2$^{QTY}$, n) NU1M vs NU1M$^{QTY}$, o) NU2M vs NU2M$^{QTY}$, p) NU3M vs NU3M$^{QTY}$, q) NU4M vs NU4M$^{QTY}$, r) NU5M vs NU5M$^{QTY}$, s) NU6M vs NU6M$^{QTY}$, and t) NU4LM vs NU4LM$^{QTY}$.

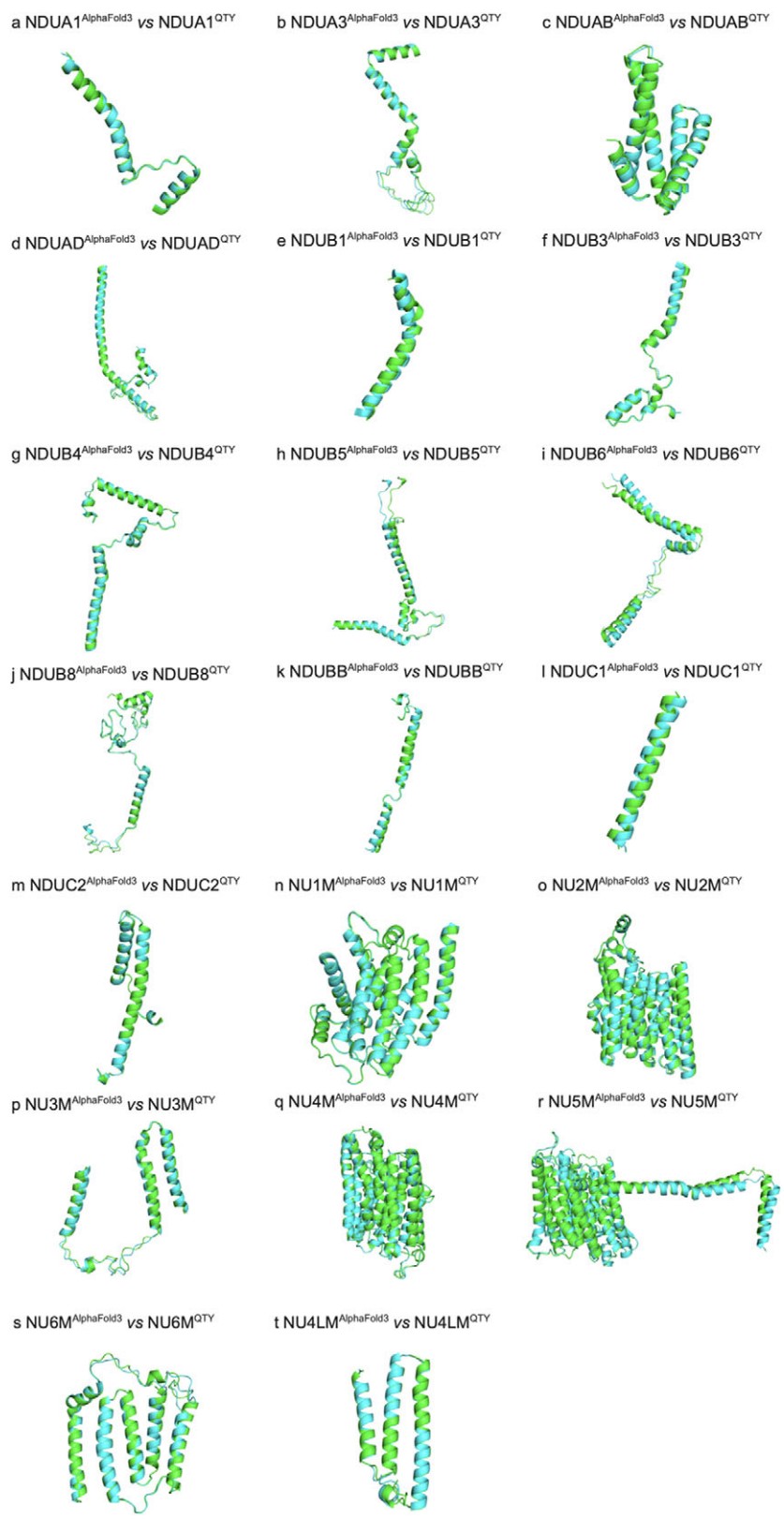

**Figure 3. Superpositions of AlphaFold 3-predicted structures of native and their QTY enzyme analogs**. Color code: green = AlphaFold 3-predicted native structures; cyan = AlphaFold 3-predicted water-soluble QTY analogs. a) NDUA1 vs NDUA1$^{QTY}$ (RMSD = 0.637Å), b) NDUA3 vs NDUA3$^{QTY}$ (RMSD = 0.400Å), c) NDUAB vs NDUAB$^{QTY}$ (RMSD = 0.374Å), d) NDUAD vs NDUAD$^{QTY}$ (RMSD = 0.570Å), e) NDUB1 vs NDUB1$^{QTY}$ (RMSD = 2.180Å), f) NDUB3 vs NDUB3$^{QTY}$ (RMSD = 1.110Å), g) NDUB4 vs NDUB4$^{QTY}$ (RMSD = 0.687Å), h) NDUB5 vs NDUB5$^{QTY}$ (RMSD = 0.511Å), i) NDUB6 vs NDUB6$^{QTY}$ (RMSD = 3.127Å), j) NDUB8 vs NDUB8$^{QTY}$ (RMSD = 0.773Å), k) NDUBB vs NDUBB$^{QTY}$ (RMSD = 0.478Å), l) NDUC1 vs NDUC1$^{QTY}$ (RMSD = 1.283Å), m) NDUC2 vs NDUC2$^{QTY}$ (RMSD = 0.184Å), n) NU1M vs NU1M$^{QTY}$ (RMSD = 0.308Å), o) NU2M vs NU2M$^{QTY}$ (RMSD = 0.390Å), p) NU3M vs NU3M$^{QTY}$ (RMSD = 0.837Å), q) NU4M vs NU4M$^{QTY}$ (RMSD = 0.270Å), r) NU5M vs NU5M$^{QTY}$ (RMSD = 0.262Å), s) NU6M vs NU6M$^{QTY}$ (RMSD = 0.541Å), and t) NU4LM vs NU4LM$^{QTY}$ (RMSD = 0.528Å).

0.262Å), s) NU6M *vs* NU6M$^{QTY}$ (RMSD = 0.541Å), t) NU4LM *vs* NU4LM$^{QTY}$ (RMSD = 0.528Å).

The RMSD of NDUB1 (RMSD = 2.180Å) and NDUB6 (RMSD = 3.127Å) shows that AlphaFold 3 might not be as accurate in the prediction of these two proteins. The overall low RMSD shows that the AlphaFold 3 predicted water-soluble QTY analogs share very similar structures with their native transmembrane proteins.

### Superpositions of CryoEM structures with AlphaFold 3-predicted native transmembrane enzymes and their water-soluble QTY analogs

To combine the CryoEM-determined native structures, AlphaFold 3-predicted native proteins, and AlphaFold 3-predicted QTY analogs, we superpose all three structures together to get a holistic view of how similar these structures are. The three different kinds of structures superposed very well (Figure 4). The superposed structures all seem reasonable and superposed well.

### Analysis of the hydrophobic surface of native transmembrane enzymes and their water-soluble QTY analogs

To study these hydrophobic transmembrane enzymes, they need to be separated from their lipid bilayer membranes using detergents, which disrupt the interactions between the membrane enzymes and solubilize the transmembrane proteins. Without proper detergent for isolation, the hydrophobic nature of these enzymes causes them to aggregate and precipitate, leading to a loss in biological function.

The hydrophobic surfaces are represented in yellowish patches (Figure 5). For clarity of view, the extramembrane region is disregarded to clearly view the changes in the hydrophobic patches originating from the transmembrane domains of the proteins. The transmembrane domains are embedded within the hydrophobic lipid bilayer, where nonpolar and hydrophobic amino acids including Leucine (L), Isoleucine (I), Valine (V), Phenylalanine (F), Methionine (M), Tryptophan (W), and Alanine (A) exclude water by interacting with lipid molecules.

After applying the QTY code to replace the hydrophobic amino acids L, I/V, and F, with hydrophilic amino acids glutamine (Q), threonine (T), and tyrosine (Y), the hydrophobic surface areas are significantly reduced. More importantly, since the electron density of the amino acids replaced are similar, the alpha-helix structure of the QTY analogs retained its structural integrity and stability, an observation that is consistent with previous experiments performed on chemokine, cytokine receptors, and bacterial histidine kinase (Zhang et al., 2018; Hao et al., 2020; Li et al., 2024).

### DockQ score of AlphaFold 3-predicted water-soluble QTY analog megacomplex

The DockQ score shows the quality of an interface of a model compared with the native structure, which combines the fraction of native contacts (Fnat), ligand root mean square deviation (LRMS), and interface root mean square deviation (iRMS) standardized by the CAPRI criteria to produce a score from 0 to 1 (Basu and Wallner, 2016). The DockQ can be used to evaluate the quality of protein docking models, where a value exceeding 0.80 implies high accuracy, between 0.80 and 0.49 medium accuracy, and between 0.49 and 0.23 acceptable accuracy (Zhu et al., 2023).

The overall DockQ score for the native CI complex and its AlphaFold 3 predicted QTY-analog complex yielded a score of 0.712, which suggests a medium-quality docking. Additionally,

DockQ analyzed the 49 interfaces, which produced a median DockQ score of 0.731, confirming the medium to high quality of the prediction. The median Fnat of 0.5 suggests that approximately half of the native contacts are preserved in the QTY-analog structure. The median LRMS of 1.965Å and median iRMS of 0.905Å demonstrate the ligand's overall alignment with the native structure and highly accurate interface alignment, respectively. These results suggest that the QTY-analog of CI retains a high degree of structural fidelity to the native complex (Table 3).

### Superpositions of CryoEM megacomplex structures with AlphaFold 3-predicted native transmembrane enzymes and their water-soluble QTY analog megacomplex

The individual enzymes in the mitochondrial Complex CI are shown to be apt for QTY substitution, with their QTY analogs showing high structural similarities to their native forms. We ask whether these proteins will maintain their original interactions to form a similar complex after applying the QTY code.

We first used AlphaFold 3 to predict the mitochondrial Complex CI, which contains twenty membrane proteins. Then, we superposed the CryoEM-determined native structure with their QTY analog (Figure 6). The complex superposed well (RMSD = 1.647Å). The high structural similarity not only shows AlphaFold 3's capability in predicting protein–protein interactions well but also indicates the feasibility of applying the QTY substitution systematically to an entire complex while still maintaining its original function.

### AlphaFold 3 predictions

DeepMind released AlphaFold 3 in May 2024, marking a significant leap in accuracy for modeling across biomolecular space. This latest iteration outperforms state-of-the-art docking tools and its predecessor, AlphaFold-Multimer v.2.3, in protein structure and protein–protein interaction predictions (Abramson et al., 2024). AlphaFold 3 reduces the reliance on multiple sequence alignment by integrating a diffusion-based model, enabling it to predict a broader spectrum of biomolecules, including ligands, ions, nucleic acids, modified residues, and large protein megacomplexes. On October 9, 2024, the Nobel Prize in Chemistry was awarded to DeepMind's founders, Demis Hassabis and John Jumper, for their contribution in revolutionizing how computation machine learning/AI advance structural biology.

AlphaFold 3 is easily accessible online (https://alphafoldserver.com), allowing users to make 20 predictions a day. The structures of the QTY analogs were predicted using the AlphaFold 3 server, which was run free of charge and the results were produced within a few minutes.

DeepMind also collaborates with the EBI to make over 214 million predicted protein structures available through the AlphaFold Protein Structure Database (https://alphafold.ebi.ac.uk). This number is continuously expanding, with the quality of predictions improving further with the advent of AlphaFold 3.

Despite its advancements, AlphaFold 3 still has limitations, which we encountered in our study. One major constraint is its ability to predict structures with a maximum length of 5,000 residues. While our initial plan was to analyze the entire mitochondrial complexes CI, CII, and CIV, we quickly realized that the AlphaFold server could not process such large and intricate structures. Even for complexes within the 5,000-residue limit, AlphaFold 3 occasionally fails to generate predictions. Fortunately, mitochondrial complex CI fell within this threshold, allowing us to leverage AlphaFold 3's capabilities successfully.

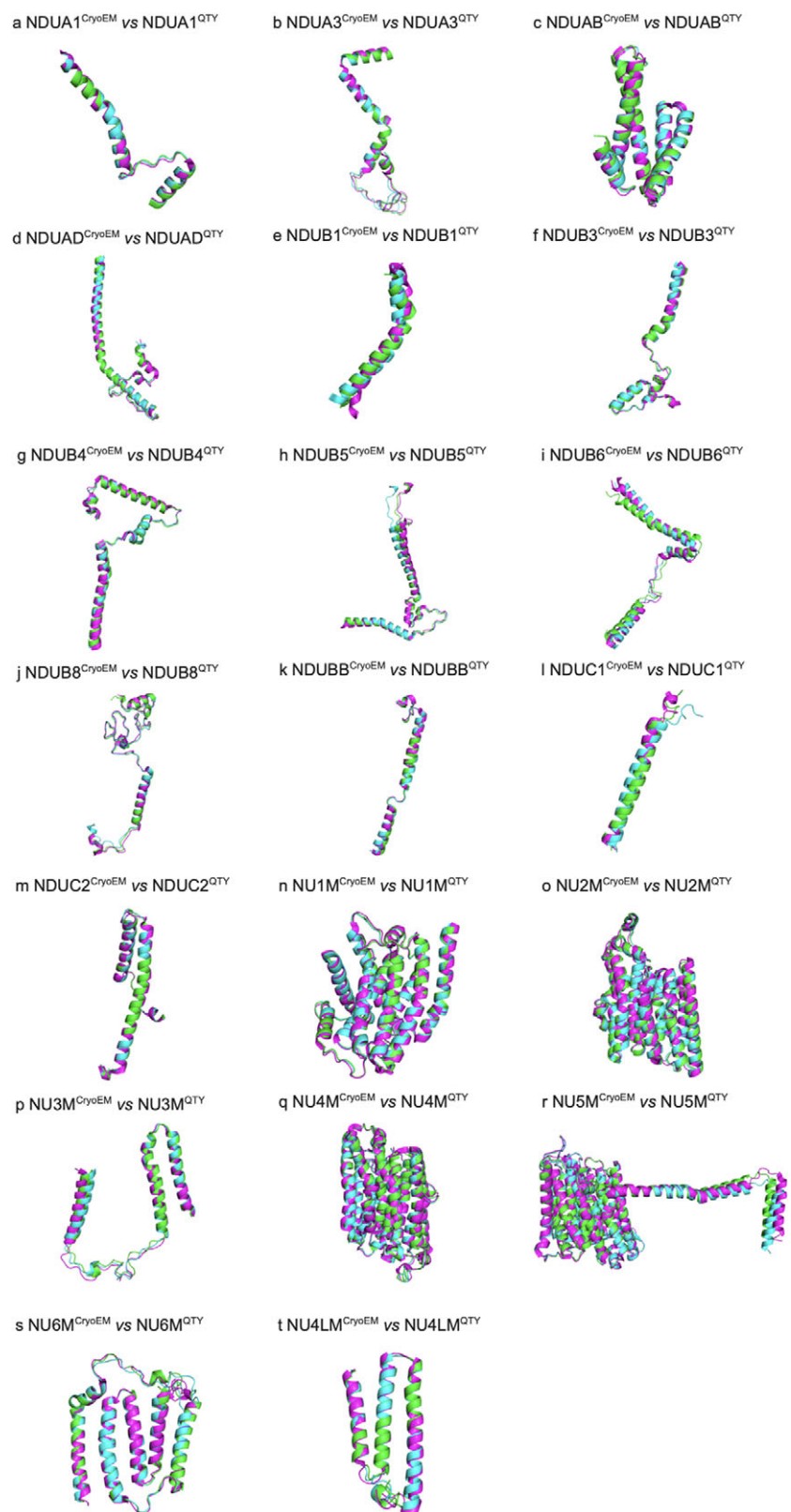

**Figure 4. Superpositions of CryoEM structures with AlphaFold 3-predicted native integral membrane enzymes and their water-soluble QTY analogs**. Superposition of i) the experimentally determined CryoEM structures (magenta) with ii) AlphaFold 3-predicted structures (green) and iii) AlphaFold 3-predicted water-soluble QTY analog structures (cyan). These superpositions are shown in Figure 4. These three different kinds of structures are apparently superposed very well. The differences and variations are insignificant. a) NDUA1$^{CryoEM}$/NDUA1$^{Native}$/NDUA1$^{QTY}$, b) NDUA3$^{CryoEM}$/NDUA3$^{Native}$/NDUA3$^{QTY}$, c) NDUAB$^{CryoEM}$/NDUAB$^{Native}$/NDUAB$^{QTY}$, d) NDUAD$^{CryoEM}$/NDUAD$^{Native}$/NDUAD$^{QTY}$, e) NDUB1$^{CryoEM}$/ NDUB1$^{Native}$/NDUB1$^{QTY}$, f) NDUB3$^{CryoEM}$/NDUB3$^{Native}$/NDUB3$^{QTY}$, g) NDUB4$^{CryoEM}$/NDUB4$^{Native}$/NDUB4$^{QTY}$, h) NDUB5$^{CryoEM}$/NDUB5$^{Native}$/NDUB5$^{QTY}$, i) NDUB6$^{CryoEM}$/NDUB6$^{Native}$/ NDUB6$^{QTY}$, j) NDUB8$^{CryoEM}$/NDUB8$^{Native}$/NDUB8$^{QTY}$, k) NDUBB$^{CryoEM}$/NDUBB$^{Native}$/NDUBB$^{QTY}$, l) NDUC1$^{CryoEM}$/NDUC1$^{Native}$/NDUC1$^{QTY}$, m) NDUC2$^{CryoEM}$/NDUC2$^{Native}$/NDUC2$^{QTY}$, n) NU1M$^{CryoEM}$/NU1M$^{Native}$/NU1M$^{QTY}$, o) NU2M$^{CryoEM}$/NU2M$^{Native}$/NU2M$^{QTY}$, p) NU3M$^{CryoEM}$/NU3M$^{Native}$/NU3M$^{QTY}$, q) NU4M$^{CryoEM}$/NU4M$^{Native}$/NU4M$^{QTY}$, r) NU5M$^{CryoEM}$/NU5M$^{Native}$/ NU5M$^{QTY}$, s) NU6M$^{CryoEM}$/NU6M$^{Native}$/NU6M$^{QTY}$, and t) NU4LM$^{CryoEM}$/NU4LM$^{Native}$/NU4LM$^{QTY}$.

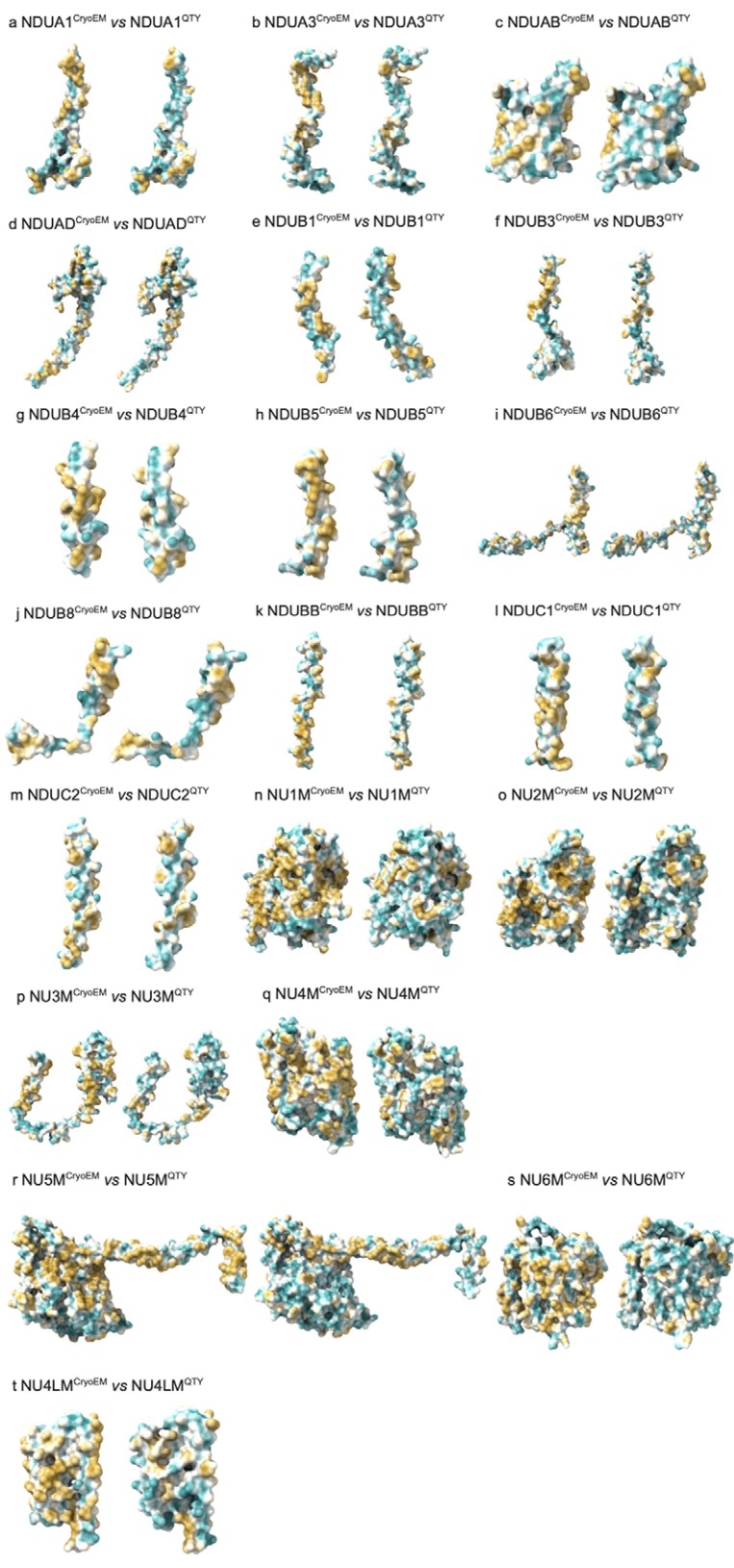

**Figure 5. Hydrophobic surface of six integral membrane enzymes and their water-soluble QTY analogs**. The native proteins have many hydrophobic residues L, I, V, and F in the transmembrane helices. After Q, T, and Y substitutions of L, I and V, and F respectively, the hydrophobic surface patches (yellowish) in the transmembrane helices become more hydrophilic (cyan). For clarity of direct comparisons, unstructured loops in the CryoEM structures were removed in the QTY analogs. a) NDUA1 vs NDUA1$^{QTY}$, b) NDUA3 vs NDUA3$^{QTY}$, c) NDUAB vs NDUAB$^{QTY}$, d) NDUAD vs NDUAD$^{QTY}$, e) NDUB1 vs NDUB1$^{QTY}$, f) NDUB3 vs NDUB3$^{QTY}$, g) NDUB4 vs NDUB4$^{QTY}$, h) NDUB5 vs NDUB5$^{QTY}$, i) NDUB6 vs NDUB6$^{QTY}$, j) NDUB8 vs NDUB8$^{QTY}$, k) NDUBB vs NDUBB$^{QTY}$, l) NDUC1 vs NDUC1$^{QTY}$, m) NDUC2 vs NDUC2$^{QTY}$, n) NU1M vs NU1M$^{QTY}$, o) NU2M vs NU2M$^{QTY}$, p) NU3M vs NU3M$^{QTY}$, q) NU4M vs NU4M$^{QTY}$, r) NU5M vs NU5M$^{QTY}$, s) NU6M vs NU6M$^{QTY}$, and t) NU4LM vs NU4LM$^{QTY}$.

**Table 3.** The DockQ score of QTY analog of Mitochondrial Complex I and 49 native interfaces

| Native Chains | QTY Chains | DockQ | Fnat | LRMS | iRMS |
|---|---|---|---|---|---|
| CI$^{CryoEM}$ | CI$^{QTY}$ | 0.712 | – | – | – |
| NDUA1$^{CryoEM}$, NDUAD$^{CryoEM}$ | NDUA1$^{QTY}$, NDUAD$^{QTY}$ | 0.876 | 0.840 | 1.282 | 0.729 |
| NDUA1$^{CryoEM}$, NU6M$^{CryoEM}$ | NDUA1$^{QTY}$, NU6M$^{QTY}$ | 0.444 | 0.375 | 5.906 | 2.399 |
| NDUA1$^{CryoEM}$, NU1M$^{CryoEM}$ | NDUA1$^{QTY}$, NU1M$^{QTY}$ | 0.731 | 0.500 | 1.197 | 0.952 |
| NDUA3$^{CryoEM}$, NDUAD$^{CryoEM}$ | NDUA1$^{QTY}$, NDUAD$^{QTY}$ | 0.862 | 0.667 | 1.430 | 0.357 |
| NDUA3$^{CryoEM}$, NU3M$^{CryoEM}$ | NDUA3$^{QTY}$, NU3M$^{QTY}$ | 0.726 | 0.438 | 2.354 | 0.726 |
| NDUA3$^{CryoEM}$, NU6M$^{CryoEM}$ | NDUA3$^{QTY}$, NU6M$^{QTY}$ | 0.580 | 0.000 | 4.893 | 0.168 |
| NDUA3$^{CryoEM}$, NU1M$^{CryoEM}$ | NDUA3$^{QTY}$, NU1M$^{QTY}$ | 0.712 | 0.457 | 2.056 | 0.900 |
| NDUAB$^{CryoEM}$, NDUB5$^{CryoEM}$ | NDUAB$^{QTY}$, NDUB5$^{QTY}$ | 0.914 | 1.000 | 1.164 | 0.843 |
| NDUAB$^{CryoEM}$, NU2M$^{CryoEM}$ | NDUAB$^{QTY}$, NU2M$^{QTY}$ | 0.777 | 0.606 | 1.308 | 0.869 |
| NDUAB$^{CryoEM}$, NU4LM$^{CryoEM}$ | NDUAB$^{QTY}$, NU4LM$^{QTY}$ | 0.448 | 0.000 | 2.258 | 1.804 |
| NDUAB$^{CryoEM}$, NU5M$^{CryoEM}$ | NDUAB$^{QTY}$, NU5M$^{QTY}$ | 0.616 | 0.227 | 0.879 | 1.148 |
| NDUAB$^{CryoEM}$, NU4M$^{CryoEM}$ | NDUAB$^{QTY}$, NU4M$^{QTY}$ | 0.910 | 1.000 | 0.939 | 0.885 |
| NDUAD$^{CryoEM}$, NU3M$^{CryoEM}$ | NDUAD$^{QTY}$, NU3M$^{QTY}$ | 0.784 | 0.800 | 2.241 | 1.182 |
| NDUAD$^{CryoEM}$, NU6M$^{CryoEM}$ | NDUAD$^{QTY}$, NU6M$^{QTY}$ | 0.326 | 0.167 | 4.860 | 6.037 |
| NDUAD$^{CryoEM}$, NU1M$^{CryoEM}$ | NDUAD$^{QTY}$, NU1M$^{QTY}$ | 0.749 | 0.615 | 2.853 | 0.905 |
| NDUB3$^{CryoEM}$, NU5M$^{CryoEM}$ | NDUB3$^{QTY}$, NU5M$^{QTY}$ | 0.695 | 0.523 | 2.013 | 1.182 |
| NDUB5$^{CryoEM}$, NDUB6$^{CryoEM}$ | NDUB5$^{QTY}$, NDUB6$^{QTY}$ | 0.771 | 0.857 | 2.483 | 1.401 |
| NDUB5$^{CryoEM}$, NDUBB$^{CryoEM}$ | NDUB5$^{QTY}$, NDUBB$^{QTY}$ | 0.734 | 0.385 | 1.017 | 0.676 |
| NDUB5$^{CryoEM}$, NDUC2$^{CryoEM}$ | NDUB5$^{QTY}$, NDUC2$^{QTY}$ | 0.940 | 0.947 | 1.188 | 0.526 |
| NDUB5$^{CryoEM}$, NU2M$^{CryoEM}$ | NDUB5$^{QTY}$, NU2M$^{QTY}$ | 0.879 | 1.000 | 1.723 | 1.039 |
| NDUB5$^{CryoEM}$, NU5M$^{CryoEM}$ | NDUB5$^{QTY}$, NU5M$^{QTY}$ | 0.898 | 1.000 | 1.405 | 0.934 |
| NDUB5$^{CryoEM}$, NDUB1$^{CryoEM}$ | NDUB5$^{QTY}$, NDUB1$^{QTY}$ | 0.775 | 0.500 | 2.574 | 0.475 |
| NDUB5$^{CryoEM}$, NU4M$^{CryoEM}$ | NDUB5$^{QTY}$, NU4M$^{QTY}$ | 0.830 | 0.761 | 1.122 | 0.879 |
| NDUB6$^{CryoEM}$, NU5M$^{CryoEM}$ | NDUB6$^{QTY}$, NU5M$^{QTY}$ | 0.764 | 0.653 | 2.842 | 0.887 |
| NDUB8$^{CryoEM}$, NU5M$^{CryoEM}$ | NDUB8$^{QTY}$, NU5M$^{QTY}$ | 0.847 | 0.765 | 1.114 | 0.768 |
| NDUB8$^{CryoEM}$, NDUB4$^{CryoEM}$ | NDUB8$^{QTY}$, NDUB4$^{QTY}$ | 0.920 | 0.946 | 0.865 | 0.696 |
| NDUB8$^{CryoEM}$, NU4M$^{CryoEM}$ | NDUB8$^{QTY}$, NU4M$^{QTY}$ | 0.811 | 0.688 | 1.386 | 0.815 |
| NDUBB$^{CryoEM}$, NU5M$^{CryoEM}$ | NDUBB$^{QTY}$, NU5M$^{QTY}$ | 0.717 | 0.500 | 1.376 | 1.035 |
| NDUBB$^{CryoEM}$, NU4M$^{CryoEM}$ | NDUBB$^{QTY}$, NU4M$^{QTY}$ | 0.689 | 0.274 | 1.080 | 0.733 |
| NDUC1$^{CryoEM}$, NDUC2$^{CryoEM}$ | NDUC1$^{QTY}$, NDUC2$^{QTY}$ | 0.700 | 0.227 | 0.983 | 0.542 |
| NDUC2$^{CryoEM}$, NU2M$^{CryoEM}$ | NDUC2$^{QTY}$, NU2M$^{QTY}$ | 0.707 | 0.480 | 1.227 | 1.074 |
| NDUC2$^{CryoEM}$, NDUB1$^{CryoEM}$ | NDUC2$^{QTY}$, NDUB1$^{QTY}$ | 0.556 | 0.000 | 3.105 | 0.783 |
| NDUC2$^{CryoEM}$, NDUB4$^{CryoEM}$ | NDUC2$^{QTY}$, NDUB4$^{QTY}$ | 0.899 | 1.000 | 2.460 | 0.810 |
| NDUC2$^{CryoEM}$, NU4M$^{CryoEM}$ | NDUC2$^{QTY}$, NU4M$^{QTY}$ | 0.858 | 0.900 | 1.567 | 0.967 |
| NU2M$^{CryoEM}$, NU3M$^{CryoEM}$ | NU2M$^{QTY}$, NU3M$^{QTY}$ | 0.890 | 1.000 | 3.081 | 0.785 |
| NU2M$^{CryoEM}$, NU4LM$^{CryoEM}$ | NU2M$^{QTY}$, NU4LM$^{QTY}$ | 0.581 | 0.271 | 1.954 | 1.434 |
| NU2M$^{CryoEM}$, NU5M$^{CryoEM}$ | NU2M$^{QTY}$, NU5M$^{QTY}$ | 0.671 | 0.432 | 2.069 | 1.132 |
| NU2M$^{CryoEM}$, NU6M$^{CryoEM}$ | NU2M$^{QTY}$, NU6M$^{QTY}$ | 0.443 | 0.433 | 6.370 | 2.561 |
| NU2M$^{CryoEM}$, NU4M$^{CryoEM}$ | NU2M$^{QTY}$, NU4M$^{QTY}$ | 0.670 | 0.365 | 2.052 | 0.984 |
| NU3M$^{CryoEM}$, NU4LM$^{CryoEM}$ | NU3M$^{QTY}$, NU4LM$^{QTY}$ | 0.514 | 0.095 | 2.196 | 1.470 |
| NU3M$^{CryoEM}$, NU6M$^{CryoEM}$ | NU3M$^{QTY}$, NU6M$^{QTY}$ | 0.564 | 0.340 | 2.930 | 1.632 |
| NU3M$^{CryoEM}$, NU1M$^{CryoEM}$ | NU3M$^{QTY}$, NU1M$^{QTY}$ | 0.620 | 0.360 | 1.965 | 1.357 |

**Table 3** *Continued*

| Native Chains | QTY Chains | DockQ | Fnat | LRMS | iRMS |
|---|---|---|---|---|---|
| NU4LM[CryoEM], NU5M[CryoEM] | NU4LM[QTY], NU5M[QTY] | 0.466 | 0.222 | 3.010 | 2.366 |
| NU4LM[CryoEM], NU6M[CryoEM] | NU4LM[QTY], NU6M[QTY] | 0.425 | 0.256 | 2.784 | 4.165 |
| NU5M[CryoEM], NDUB4[CryoEM] | NU5M[QTY], NDUB4[QTY] | 0.865 | 0.824 | 1.221 | 0.771 |
| NU5M[CryoEM], NU4M[CryoEM] | NU5M[QTY], NU4M[QTY] | 0.763 | 0.492 | 1.091 | 0.718 |
| NU6M[CryoEM], NU1M[CryoEM] | NU6M[QTY], NU1M[QTY] | 0.503 | 0.222 | 6.667 | 1.058 |
| NDUB1[CryoEM], NU4M[CryoEM] | NDUB1[QTY], NU4M[QTY] | 0.677 | 0.409 | 2.153 | 1.027 |
| NDUB4[CryoEM], NU4M[CryoEM] | NDUB4[QTY], NU4M[QTY] | 0.798 | 0.656 | 1.463 | 0.825 |

*Note:* The evaluation included 50 interface regions, yielding a **DockQ score** of **0.712** when comparing the native CI structure and its QTY-analog as predicted by AlphaFold 3, which indicates medium quality $(0.49 \leq DockQ < 0.80)$ docking. The median of DockQ score for the 49 additional interface is 0.731, Fnat is 0.5, LRMS is 1.965Å, and iRMS is 0.905Å. These results suggest that the QTY-analog retains a high degree of structural fidelity to the native complex.
Abbreviations: Fnat, fraction of native contacts; LRMS, ligand root mean square deviation; iRMS, interface root mean square deviation; and –, not applicable.

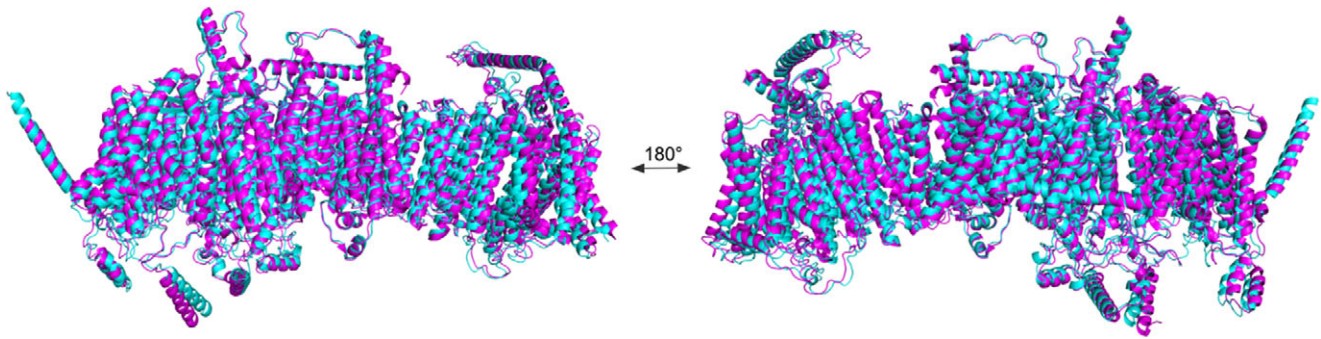

**Figure 6. Superpositions of CryoEM-determined structures of mitochondrial transmembrane Complex I megacomplex and its AlphaFold 3-predicted water-soluble QTY analogs**. The CryoEM-determined structures of the mitochondrial complex are obtained from the Protein Data Bank (PDB). The CryoEM structure (magenta) is superposed with its QTY analog (cyan) predicted by AlphaFold 3. These superposed structures show that the membrane complex and its QTY analog have very similar structures (RMSD = 1.601Å). For clarity of direct comparisons, unstructured loops in the CryoEM structure were removed in the QTY analogs.

### The integral transmembrane protein megacomplex in this study

In this study, using the advanced capability of AlphaFold 3, we extended the QTY code to megacomplex protein structures and investigated whether the resulting QTY analogs retain their native protein–protein interactions. The mitochondrial Complex I selected for this study is critical in the electron transport and ATP production in the heart, skeletal muscle, brain, liver, and kidney. By reducing the hydrophobicity of this complex, we hope to gain deeper insights into the highly efficient coupling of electron transfer and proton pumping, as well as the associated conformational changes within the megacomplex.

The water-soluble QTY analogs generated in this study hold significant potential: (i) they may help validate and generalize the QTY code system for more intricate protein assemblies, ii) some of these individual QTY code-engineered membrane protein analogs in this megacomplex could be used as water-soluble antigens to generate therapeutic mAbs, and iii) the mitochondrial Complex I could also serve as promising therapeutic targets for the treatment of various neurodegenerative diseases.

### Conclusion

Proteins can generally be classified into two groups: Class I (hydrophilic) and Class II (hydrophobic) (Branden and Tooze, 1999; Fersht, 2017; Zhang and Egli, 2022). More specifically, proteins often consist of three analogs of alpha-helices: i) **Type I**, composed of hydrophilic amino acids (D, E, H, N, Q, K, R, S, T, and Y), which are commonly found in water-soluble globular proteins; ii) **Type II**, composed of hydrophobic amino acids (L, I, V, F, M, A, W, and P), typically located in the helical transmembrane regions of membrane proteins; and iii) **Type III**, amphiphilic helices, containing nearly equal proportions of hydrophilic and hydrophobic amino acids that partition into distinct hydrophobic and hydrophilic faces (Branden and Tooze, 1999; Fersht, 2017). Inspired by the exceptional water solubility of hemoglobin, a protein predominantly composed of alpha-helices, we developed the QTY code to systematically replace hydrophobic α-helices with hydrophilic ones. This approach leverages insights from high-resolution (1.5Å) electron density maps of 20 amino acids, which revealed structural similarities between hydrophobic and hydrophilic amino acid pairs: leucine (L) to glutamine (Q), isoleucine (I)/valine (V) to threonine (T), and phenylalanine (F) to tyrosine (Y).

Thus far, because of AlphaFold 3's recent release, very few protein megacomplexes have been studied using AlphaFold 3. In this study, we applied the QTY code to mitochondrial Complex I to engineer water-soluble QTY analogs. To evaluate the structural impact of these modifications, we used AlphaFold 3 to predict the structures of the QTY analogs and superimposed them onto their respective native protein structures. Additionally, we employed a suite of in silico computational and bioinformatic tools to analyze sequence and structural features related to protein stability and water solubility. Our findings demonstrated that the QTY code effectively reduced the hydrophobic surfaces of the proteins while maintaining

high structural similarity between the QTY analogs and their native counterparts. Furthermore, the QTY analogs retained their structural integrity, as they successfully assembled into a megacomplex I structure comparable to the CryoEM-determined native megacomplex. These hydrophilic proteins can now be used as water-soluble antigens for the discovery of therapeutic mAbs for the treatment of a wide range of neurodegenerative diseases.

## Methods

### Protein sequence alignments and other characteristics

The native protein sequences for NDUA1, NDUA3, NDUAB, NDUAD, NDUB1, NDUB3, NDUB4, NDUB5, NDUB6, NDUB8, NDUBB, NDUC1, NDUC2, NU1M, NU2M, NU3M, NU4M, NU5M, NU6M, and NU4LM. were obtained from UniProt (https://www.uniprot.org). The sequences for the QTY analogs were aligned using the same methods as previously described. The MWs and pI values of the proteins were calculated using the Expasy (https://web.expasy.org/compute_pi/)

### AlphaFold 3 predictions

The protein structures of the QTY analogs were predicted using the AlphaFold 3 server (https://alphafoldserver.com/). PBD files for the predicted native protein structures were obtained from The EBI (https://alphafold.ebi.ac.uk), which contains all AlphaFold 3 predicted structures for native proteins. The UniProt website (https://www.uniprot.org) provided protein ID, entry name, description, and FASTA sequence for each native protein. The QTY code can be applied to FASTA sequences through the QTY method website (https://pss.sjtu.edu.cn/). The website also provides MWs, pI values, TM variation, and overall variation.

### Superposed structures

PBD files for native protein structures experimentally determined by CryoEM were taken from the PDB: 5XTC. Predictions for the QTY analogs were carried out using the AlphaFold 3 server, which can be found at https://alphafoldserver.com/. These structures were superposed using the PyMOL "super" command and the RMSDs were calculated based on Ca atoms (https://pymol.org). For simplicity and clarity, unstructured loops and extraneous protein monomers were removed from the figures.

### Structure visualization

PyMOL (https://pymol.org) was used to superpose the native protein structure and the QTY analog. UCSF Chimera (https://www.rbvi.ucsf.edu/chimera) was used to render each protein model with hydrophobicity patches.

### Docking evaluation

DockQ (http://github.com/bjornwallner/DockQ/) was used to assess the quality of protein docking models of the QTY analog of Mitochondrial Complex I.

### Data availability of AlphaFold 3-predicted water-soluble QTY analogs.

EBI (https://alphafold.ebi.ac.uk) serves as a database that provides open access to more than 214 million AlphaFold 3-predicted protein structures. Protein characteristics used in the analysis are available on UniProt (https://www.uniprot.org/). The native CryoEM-determined six integral membrane protein enzymes are available in the RCSB PDB repository (https://www.rcsb.org/). The QTY code-designed water-soluble analogs of the human integral membrane protein enzymes are available at https://github.com/EdwardChen777/mitochondrial_complex_I. The AlphaFold 3 predicted QTY code-designed water-soluble analogs of the 20 mitochondrial CI subunits are available at https://doi.org/10.5281/zenodo.14584403. The AlphaFold 3 predicted QTY code designed water-soluble analogs of mitochondrial CI is available at https://modelarchive.org/doi/10.5452/ma-s328f. If additional information is needed, please contact the Edward Chen at edwardchen5414@gmail.com.

**Open peer review.** To view the open peer review materials for this article, please visit http://doi.org/10.1017/qrd.2025.2.

**Supplementary material.** The supplementary material for this article can be found at http://doi.org/10.1017/qrd.2025.2.

**Author contribution.** Conceptualization: S.Z. Formal analysis: E.C. Investigation and methodology: E.C. Validation: E.C. Data curation: E.C. Writing—original draft preparation: E.C. and S.Z. Review and editing: E.C. and S.Z.

**Financial support and disclosure.** E.C. is a student in transition who is applying for a Ph.D. in computer science and bioengineering for graduate study. There is no financial support for this digital structural bioinformatic study use only free online tools.

**Competing interest.** Massachusetts Institute of Technology (MIT) filed several patent applications for the QTY code for GPCRs excluding the olfactory receptors. OH₂Laboratories licensed the technology from MIT to work on water-soluble GPCR analogs. S.Z. is an inventor of the QTY code and has a minor equity in OH₂Laboratories. S.Z. is a Scientific Advisor and has minor shares for a startup RealNose to develop a sensing device based on olfactory receptors. S.Z. founded a startup 511 Therapeutics to generate therapeutic monoclonal antibodies against solute carrier transporters to treat pancreatic cancer. S.Z. has majority equity in 511 Therapeutics. E.C. declares no competing interest.

**Ethics statement.** All methods were carried out in accordance with relevant guidelines and regulations. All experimental protocols were approved by a named institutional and licensing committee. Neither human biological samples nor human subjects were used in the study. This is a completely digital structural bioinformatic study using the publicly available AlphaFold 3 machine learning program.

**Additional statement.** (1) All methods were carried out in accordance with relevant guidelines and regulations. (2) All experimental protocols were approved by a named institutional and licensing committee. (3) Neither human biological samples nor human subjects were used in the study. This is a completely digital structural bioinformatic study using the publicly available AlphaFold 3 machine learning program.

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
