## [Reviewer Report]

This project continues a series of works by Prof. Shuguang Zhang and his colleagues, who have applied the QTY approach to design soluble variants of natural transmembrane proteins. While the QTY conversion can be considered an established method, this is the first time it’s been applied to a large membrane supercomplex. While I generally like the idea and choice of the target, I have several comments:

1) My main concern is the introduction of polar residues at the interface between the complex’s subunits. A simple, straightforward application of QTY without some interface optimization might introduce many unsaturated hydrogen bond donors and acceptors, causing extensive solvation of the protein-protein interfaces and disassembly of the complex. I suggest analyzing the predicted complex’s free hydrogen bond donors/acceptors.

Other comments are more technical:

2) For all predicted models (native and QTY-converted proteins, C1 complex), quality metrics (global pLDDT, pTM, and ipTM, where applicable) must be reported either in the main text or as a supplement. For the C1 complex (native and QTY), I’d also suggest estimating the pDockQ score (details and the code are available at https://www.nature.com/articles/s41467-022-28865-w )

3) Models in AlphaFold DB are primarily predicted using AlphaFold 2 (which can be verified in the metadata in PDBx/mmCIF files). Thus, comparing AF2 DB models with QTY AF3 models is not entirely fair. I’d suggest re-predicting native proteins with AF3.

4) The authors state, “which is an upgrade from the previous version of AlphaFold2 which could only individual proteins folding.” this is inaccurate because AF2 could predict complexes in multiple modes (with original AF2 models and later with a specialized AF2-multimer model).

5) PyMOL has multiple instruments for protein structure alignment and RMSD calculation. The methods must mention the exact tool and the type of RMSD (full-heavy-atom, Ca, etc.).

6) Predicted models (at least for the C1 complex) must be deposited to ModelArchive (https://modelarchive.org/). Other models of individual proteins can be deposited to Zendo or a similar data-sharing resource.

7) Finally, I suggest trying or at least mentioning in the text other AI-assisted solubilization approaches, such as a recent example of engineering of soluble bacteriorhodopsin with SolubleMPNN neural network by Nikolaev et al. (https://www.biorxiv.org/content/10.1101/2024.11.20.624543v1).

---

## [Reviewer Report]

The authors have pioneered the “QTY code” to convert the hydrophobic solvent-accessible surfaces of helices in membrane-bound proteins to hydrophilic residues to water-solubilise them. They have recently used AlphaFold to simulate the resultant structures and compare them with the native structures. Here, they have used the more recently released AlphaFold3 to simulate QTY engineering in protein megacomplexes. They analysed QTY analogs of the mitochondrial Complex I and compared them with the native protein structures. They found the program assembled the hydrophilic analogs into the native complex structure. They discussed the accuracy of the simulations and its limitations. The results may have practical use as the indications are that the QTY analogs could be used for screening of drugs for treatment of neurodegenerative diseases.